# Is It Time for "Time-In"?: A Pilot Test of the Child-Rearing Technique

**George W. Holden [1,*], Tricia Gower [1], Sharyl E. Wee [1], Rachel Gaspar [1] and Rose Ashraf [2,3]**

[1] Department of Psychology, Southern Methodist University, Dallas, TX 75205, USA; tgower@smu.edu (T.G.); swee@smu.edu (S.E.W.); rlgaspar@gmail.com (R.G.)

[2] Department of Psychiatry and Behavioral Sciences, Boston Children's Hospital, Boston, MA 02115, USA; rose.ashraf@childrens.harvard.edu

[3] Department of Psychiatry, Harvard Medical School, Boston, MA 02115, USA

[*] Correspondence: gholden@smu.edu

**Abstract:** Time-out, a mainstay of non-punitive discipline for over 60 years, has been criticized for isolating and distancing children from others. An alternative technique, one promoted by advocates of positive parenting practices, is labeled "time-in". This procedure is intended to help the child connect to the parent, communicate their feelings, and learn how to self-regulate. Although the technique has been advocated in the positive parenting literature since at least the 1990s, there are few empirical studies evaluating it. This pilot mixed-models study was designed as an initial test to determine whether mothers, following a brief training, would use the procedure over a two-week period, and how they would view it. Based on the daily reports as well as post-intervention interview of a small sample of mothers, the technique was evaluated as easy to use and effective. This study provides initial information about mothers' use of the technique and sets the stage for a comprehensive set of studies to rigorously test and evaluate the technique.

**Keywords:** child discipline; positive parenting; positive discipline; non-punitive discipline; time-out; time-in

## 1. Introduction

The frequency of using some type of disciplinary response to what parents perceive as misbehavior steadily increases over the child's first year [1,2]. As many as 14% of parents resort to corporal (physical) punishment by the time their infants are 12 months old [2]. By the end of a child's first year, most parents have begun engaging in some form of discipline [2,3] and the frequency of correcting a child continues to increase in toddlerhood, when children are expressing their powerful drive for autonomy [4].

Along with reasoning, negotiating, withdrawing privileges, and yelling [2], parents have long resorted to inflicting pain through corporal punishment to change their children's behavior [5]. Common forms of physical punishment include slapping, spanking, paddling, and hitting. Parents rely on physical punishment for a variety of reasons, ranging from acceptance of cultural norms to a failure to control their own negative emotions [6]. Although the use of corporal punishment is commonplace in many countries [7], physical punishment is not a benign behavior. For well over 50 years, researchers have found negative associations between corporal punishment, as well as other forms of punitive discipline, and children's behavioral, emotional, and cognitive development [8–10].

One widely promoted alternative to physical punishment is time-out. There are conflicting accounts about the origins of time-out. The psychologist Arthur Staats [11], who labeled himself as a "social behaviorist," claimed he developed the disciplinary technique that was non-punitive and withheld attention (i.e., reward) to undesirable behaviors [12]. Although the technique can be seen in paintings and found in novels at least since 1894, Staats indicated that he created it and labeled this technique *time-out*, shorthand for time-out from positive reinforcement [13]. An alternative account credits Montrose Wolf [14] with recognizing the reinforcing power of adult attention and developing the procedure [15].

Whomever developed it, it has become well known and popular. The discipline technique simply consists of removing children from reinforcing environments (e.g., attention from people and objects to play with) and placing them in less reinforcing settings. The duration of the isolation is intended to vary from 2 to 15 min, typically as a function of the child's age. In behaviorist terminology, it is a form of negative reinforcement, defined as the "contingent withdrawal of the opportunity to earn access to positive reinforcement or the loss of access to positive reinforcers for a specified time" [16].

Time-out has been a recommended form of discipline by psychologists and parent training educators for more than 50 years [17,18] and is an integral component of most evidence-based parent training programs, such as the Triple P Parenting Program [19,20]. The procedure has proved to be one of the most popular forms of discipline with parents, both in the United States and other countries [1,2,21]. For example, in Australia, it is used by between 58% and 80% of parents [19,22]. Both the American Academy of Pediatrics and the American Academy of Child and Adolescent Psychiatry recommend time-out as an effective parenting strategy. It should be recognized, however, that time-out is not a standardized procedure, rather, it can vary across eight parameters, such as whether there was a verbalized warning prior to implementation, the duration, the location, and contingent versus non-contingent release [23,24].

Since the 1960s, a considerable amount of empirical evidence has accumulated about time-outs [25–27]. From a behavioral perspective, the procedure is highly effective in modifying problematic child behavior. Yet, the procedure has also been critiqued due to its behaviorist orientation. Time-out, although non-violent, is a form of punishment and it neglects the child's emotional needs. It fails to provide a forum for communication. Importantly, by distancing the child from the parent, it does not promote warmth or dyadic connections. In fact, a number of articles have appeared both in the academic [28] and popular [29–31] literature criticizing the technique on a variety of grounds, from theoretical to neurological bases.

The foremost critique is that time-out involves the authoritarian approach of punishing children by isolating and, from children's perspective, abandoning them [28]. Another critique is that the procedure does not help the dysregulated children learn to control their emotions because the parent is not there to console, guide, and communicate [30]. Consequently, a concern, is that the technique is not helpful in promoting a positive parent–child relationship.

A contrasting theoretical orientation advocates for child-rearing actions to focus on the goal of cultivating a warm and close parent–child relationship. According to individual psychology theory [32], if parenting is oriented foremost toward promoting a positive relationship, then children will seek to please their parents and be cooperative. Unacknowledged negative emotions undermine the relationship. Consequently, the physical and emotional distance imposed by time-outs are counter-productive to promoting a warm relationship. More specifically, it is essential to have a close, warm, and positive parent–child relationship to guide children and instruct them in appropriate behavior. Adler [32] argued that children were strongly motivated to seek connections and belonging—and to feel like they are respected individuals.

Adler's child-rearing philosophy was imported to the United States and articulated further by Rudolph Dreikers [33]. Subsequently, Jane Nelsen became an early popularizer of this approach. She and her colleagues have now published a number of books and other educational materials for parents and teachers under the label of Positive Discipline [34,35]. Although the term "positive parenting" or "positive discipline" is ambiguous and ill-defined [36], there are now hundreds of books, websites, and YouTube videos advocating for the use of gentle or positive parenting techniques. In an effort to define the concept, Seay and colleagues [37] reviewed definitions from 18 articles. They definition of positive parenting is *"the continual relationship of a parent(s) and a child or children that includes caring, teaching, leading, communicating, and providing for the needs of a child consistently and unconditionally" (p. 207).* Note that the words "discipline" or "punish" are absent.

Arguably, the single most iconic example of the positive parenting orientation is *time-in*. It is a technique that is caring but also involves teaching, communicating, and providing for the emotional needs of the child. Time-in is a procedure that starkly contrasts with time-out. Instead of separating the child, time-in aims to connect to the parent with the child in a warm and loving way. During time-in, the parent sits with the child after a misdeed and helps the child calm down and regulate themself. For younger children, the procedure may involve putting the child on the parent's lap. After the child has down-regulated, the procedure involves such components as talking calmly with the child about the behavior, touching the child in a warm, supportive way (e.g., rubbing the child's back), inquiring about the child's feelings, identifying the child's needs, and informing the child how he or she could behave differently in the future. Advocates of this technique [38] theorize that it results in happier, more cooperative children.

In contrast to time-out, few researchers have studied time-in. We located two case study reports involving challenging or non-compliant children and their parents. Olmi and colleagues [39] published a study involving a $4\frac{1}{2}$-year-old boy with a severe language disability and an 8-year-old girl with a moderate mental disability. They found that the treatment package of time-in (operationalized as physical touch and verbal praise) was effective in increasing the boy's responsiveness to a teacher and dramatically reducing the frequency of throwing objects for the girl. Three years later, Mandal and colleagues [40] tested a time-in procedure with two preschoolers, referred to a psychology clinic for non-compliance. The researchers observed that the children's responsiveness and other aspects of their behavior improved dramatically after the two mothers began using the technique in the lab.

The present study represents an extension of the case studies by Mandal et al. [40] and Olmi et al. [39] by using a community sample to assess parents' willingness to use time-in and whether it could be easily taught in a brief training. Furthermore, we wanted to learn mothers' reactions to the technique including their perceptions of its efficacy. As this was a pilot study, we did not generate formal hypotheses. However, we expected that mothers would be willing to try out the technique, it could be easily taught, and, based on mothers' attitudes that logical consequences were more acceptable than mild punishments [26], they would have positive views about the technique.

## 2. Materials and Method

### 2.1. Participants

Thirty-two mothers and their 3- to 5-year-old children participated in the initial interview. These mothers had typically developing children; children with a mental diagnosis (e.g., Oppositional Defiant Disorder, Conduct Disorder, Autism Spectrum Disorder, Disruptive Mood Dysregulation Disorder, Trauma-related Disorder, and Speech Delay) were excluded and referred to local resources for family therapy. Nine mothers dropped out prior to the debriefing interview for various reasons (e.g., too busy and did not return phone calls for scheduling the second appointment). In addition, four mothers did not fill out the daily reports so they were excluded from the results. The 19 mothers with mostly complete data averaged 36 years of age (range 23–54); 37% of the target children were 3 years old, 33% were 4 years old, and 36% were 5 years old. Nine of the children were girls and nine were boys (information about the gender of one child was missing). The mothers' education level included high school graduate (11%), some college (28%), college graduate (28%), and graduate or professional degree (32%). A majority (54%) of the mothers were working either full time or part time, and the others were either homemakers (26%) or students (21%). Most (78%) of the mothers were currently married; three mothers were not married and one was divorced. Family income ranged from below USD 19,999 to more than USD 140,000, with the modal income category being between USD 100,000 and USD 119,000. Fifty-six percent of the mothers self-identified as Caucasian and the other mothers reported being Latinx (17%), Asian American (11%), African American (11%), and multi-racial (6%).

## 2.2. Measures

Parental Responses to Child Misbehavior. A 6-item version of the Parental Responses to Child Misbehavior (PRCM) [41] was used at pre- and post-intervention. Parents reported the average frequency per week that they used various disciplinary techniques over the past month (prior to the start of the study) and the past two weeks (at the debriefing interview). The six techniques they reported on were negotiate, time-in, time-out, withdraw privileges, yell in anger, and spank with hand. Two subscales were created: positive techniques (time-in and negotiate) and negative techniques (time-out, withdraw privileges, yell, and spank).

Child behavior problems. The Eyberg Child Behavior Inventory (ECBI) [42] is a commonly used short (15-item) assessment of child problems (e.g., refuses to go to bed on time and gets angry when does not get his/her own way) from the parent's perspective. Parents completed this measure before the intervention and at the follow-up interview. The frequency of problems scale uses a 7-point Likert-type scale (1 = never, 7 = always). The psychometric properties of this scale are strong [42].

Online daily survey. Parents were asked the frequency that they engaged over the course of that day in the six PRCM items (never, once, or two or more times). If the parent indicated they used time-in, a follow-up query concerned the type of child misbehavior.

Assessments of time-in. The post-intervention interview survey consisted of a series of six questions about parents' experience with and reactions to using time-in. During the interview, mothers were asked to rate each question on a 5-point Likert-type scale (1 = no or not at all, 5 = very much or many). These questions consisted of: (1) Was using time-in a positive experience? (2) How difficult was it to use? (3) How confident were you at using the procedure? (4) Did you notice changes in your child once you began using it? (5) Did you notice changes in the quality of your relationship with your child after you started using it? (6) Do you intend to use it in the future?

## 2.3. Procedure

Mothers were recruited from posters displayed in daycare centers and preschools in a large metropolitan southwestern American city (Dallas, TX). Though mothers were initially the intended sample, fathers who expressed interest were also welcomed to participate. Undergraduate research assistants went to the centers during afternoon pick-up times to describe the study and answer any questions potential participants had. Interested participants then called or emailed the lab to obtain more information and be screened for eligibility criteria. Parents were excluded from the study if they were not fluent English speakers or if children had current mental health diagnoses.

The study consisted of three segments. The first segment involved a 1.5 h visit to the lab. During that visit, mothers provided informed consent, filled out the questionnaires, and were trained in the use of time-in and an online daily discipline reporting tool. Childcare was provided by research assistants who had undergone training and were supervised. While the mothers were being trained, the assistants occupied the children with toys, coloring supplies, and movies. For added family comfort, a video-equipped baby monitor was used to allow parent and child to see each other as needed throughout the visit.

The training, developed for this study, consisted of instructing in the time-in technique. This approach is labeled a single exemplar training [43]. It was based on the psychologist Janae Weinhold's description of the seven slightly different techniques, adapted from different clinicians. All techniques were designed for children ranging in ages from 1 to 10 years old (www.weinholds.org accessed on 4 August 2018). We selected the techniques best suited for children aged 3 to 5 years of age. In general, time-in consists of observing and validating the child's emotion, inviting them to use a joint coping skill, and then discussing the issue once the child was calm. Physical affection was recommended (e.g., sit in lap, hug, and rub back). Mothers were encouraged to focus on empathy and emotion regulation, and then to set a limit or discuss a misbehavior once the child was calm.

In the training, we instructed mothers to employ three phases to the time-in: "Reflect;" "Cooling off;" and "Talk it out." The "Reflect" phase consists of helping children identify

what is upsetting them and repeating what the parent heard. The mothers were then instructed to offer their sympathy and remind the child that he or she needs to be in charge of their behavior. The children were encouraged to sit next to the mother and think about their behavior. The "Cooling off" phase of time-in, like it sounds, allows for children who engaged in rowdy or disruptive behavior, to control themselves. The parent is instructed to inform the child that their behavior was out of control or not okay and they need to sit next to the parent. The child can decide how long they needed to become quiet. The third phase, "Talk it out", involves creating physical closeness and then gently talking about the offending behavior. All of the phases are intended to teach children about emotional self-regulation while promoting a close relationship. Mothers were instructed to use their judgment about use of the phases and the fact that time-in may work better for some types of misbehaviors than others.

Instruction was provided by advanced graduate student clinicians, who were supervised by a licensed psychologist. A handout describing the steps was given to the mothers. To consolidate the training, mothers taught back the procedure to the clinician.

Following the time-in training, the assistant brought the child in to join the parent and the clinician. The parent was encouraged to teach the child about time-in, with clinician support as needed. Parents then practiced using time-in approximately 2–3 times with their children. In order to create a situation where time-in could be practiced, a "clean-up situation" was created, which is a commonly used laboratory task developed for a parent training program called Parent–Child Interaction Therapy. In this five-minute situation, the parent is asked to have the child clean up toys by him/herself. It is designed to elicit child non-compliance and resistance, and thus, is well suited to train parents in time-in. Mothers, if needed, were instructed by clinicians about the correct use of time-in. To help the child understand the technique, in a supplementary practice session, the child was encouraged to request "time-in" if they thought their parent was becoming upset.

The second segment of the study consisted of a two-week reporting period following the time-in instructional session. For 14 days, mothers received an emailed link to a short questionnaire, which was accessible via smartphone, tablet, or computer. They were asked to fill out the daily report at the end of every day. If mothers missed one or more daily reports, researchers contacted mothers by email or phone with reminders.

The third and final segment involved having the mothers return to the lab for follow-up questionnaires and a debriefing interview lasting approximately 30 min. Mothers were interviewed by a different research assistant, one who had not been involved in the initial meeting to avoid possible demand characteristics. Interviews were intended to be open-ended and conversational to enhance parent comfort in providing genuine feedback about their experiences with and evaluation of time-in.

All participants received remuneration for their time. Mothers earned USD 25 for attending the training and up to USD 35 for the daily reports and follow-up interview. Payment was provided in the form of gift cards.

## 3. Results

The vast majority of mothers (83%) who participated in the initial interview continued to participate in the study by filling out most, if not all, of the online daily reports. Nineteen of the mothers completed all or all but one of the 14 short surveys, one mother filled it out 12 times and the remaining four mothers completed 11 daily entries.

Mothers reported using time-in an average of once per day over a two-week period ($M = 8.74$). In contrast, over the two weeks, they rarely reported using time-out ($M = 0.47$). However, prior to starting the study, mothers indicated they used time-out an average of 1.68 times per day. This change reflected a significant decline, $t(18) = 6.17$, $p < 0.001$. The descriptive results and significant tests for four central pre vs. post comparisons can be found in Table 1.

According to the mothers, time-in rapidly became one of their most frequently used techniques. Paired-samples *t*-tests indicated that, over the course of two weeks of daily

diaries, participants used time-in more frequently ($M = 8.74$, $SD = 3.19$) than they used time-out ($M = 0.47$, $SD = 0.84$), $t(17) = 10.87$, $p < 0.001$, spanking ($M = 0.21$, $SD = 0.71$), $t(17) = 11.30$, $p < 0.001$, withdrawal, ($M = 2.47$, $SD = 2.59$), $t(17) = 7.68$, $p < 0.001$, and yelling, ($M = 3.32$, $SD = 2.38$), $t(17) = 7.81$, $p < 0.001$. Participants did not use time-in more frequently than negotiation, ($M = 7.32$, $SD = 3.73$), $t(17) = 1.43$, $p = 0.17$.

**Table 1.** Mothers' reported behavior, pre- and post-training.

| Variable | Pre-Training Reports | Post-Training Reports | Significance Test |
|---|---|---|---|
| Use of time-out (pre-training data = PRCM; post-training = diary reports) | $M = 1.68$ ($SD = 1.06$, range 1–5) | $M = 0.47$ ($SD = 3.10$, range 3–14) | $t(18) = 6.17$, $p < 0.001$ |
| Positive techniques (PRCM) | $M = 13.67$ ($SD = 3.74$, range 6–21) | $M = 14.72$ ($SD = 3.59$, range 7–21) | $t(17) = 1.36$, $p < 0.01$ |
| Negative techniques (PRCM) | $M = 23.06$ ($SD = 5.11$, range 15–33) | $M = 22.56$ ($SD = 5.63$, range 15–32) | $t(17) = 0.40$, $p = 0.70$ |

Notes. $N = 19$.

The use of time-in also appeared to result in some benign collateral repercussions. Pre- and post-training reports on the PRCM indicated mothers shifted their disciplinary responses to their children's misdeeds toward more positive techniques. When subscales of positive and negative disciplinary techniques were formed, they revealed that over the two weeks of online reporting, mothers reported using positive parenting techniques significantly more frequently ($M = 8.03$, $SD = 0.62$) than negative parenting techniques ($M = 1.59$, $SD = 0.37$), $t(17) = 11.65$, $p < 0.001$.

A similar result can be seen when comparing maternal reports on the PRCM about their disciplinary responses before and after the training. Prior to training, mothers indicated they used positive techniques an average of 13.67 per week ($SD = 3.74$), but negative parenting techniques an average of 23.06 ($SD = 5.62$). Following the training, mothers reported using positive techniques an average of 14.72 ($SD = 3.59$) and negative parenting techniques a mean of 22.56 ($SD = 5.63$). Those reports indicated a modest significant increase in the reported rate of using positive techniques $t(17) = 1.36$, $p < 0.001$, but no difference in the rate of using negative techniques, $t(17) = 0.40$, $p = 0.70$.

Reports of child behavior problems on the ECBI indicated a modest but non-significant decrease in problems over the two weeks. With regard to frequency of problems, at the follow-up, mothers identified fewer problems than at the start of the study, $Ms = 107.4$ and 102.7, respectively, $t[17] = 1.053$, $p < 0.30$.

The debriefing interviews revealed the mothers' enthusiasm for the procedure. Despite the brief training, on average, the mothers disagreed ($M = 2.11$) with the statement "I found time-in difficult to use." In fact, they were very confident ($M = 4.37$) in using the technique. The means, standard deviations, and ranges for the six key debriefing questions can be found in Table 2. They were also favorably inclined about the technique after using it for just two weeks. One mother exclaimed "Most effective thing [teaching technique] I've ever done!" The parents recognized that the technique modeled how to calm down and promoted communication from both the adult perspective (i.e., the child was able to better understand their misdeed) and the child's perspective (time-in provided a way for the children to communicate about their needs and desires). On average, the mothers agreed ($M = 4.00$) with the statement that they had observed changes in their child since using time-in.

One unexpected observation made by some of the mothers was that using the technique prompted them to reflect on their parenting. In the debriefing interview, one mother mentioned that the procedure reminded her of the menu of options she can use to address child behavioral problems. More generally, another mother recognized "I've become more aware of my parenting—like looking for my child's cues, taking more little breaks throughout the day." Furthermore, most of the mothers reported noticing changes ($M = 3.90$) in

how they interact with their child beyond the use of time-in. The mothers also recognized that the use of time-in promoted a sense of connection with their children. As a mother reported, "it helped me see my child more as an equal." Perhaps most telling, many mothers expressed liking the technique and, on average, strongly agreed (*M* = 4.60) with the statement "I intend to continue to use time-in."

**Table 2.** Mothers' views about time-in after using it for two weeks.

| Question | *M* | *SD* | Range |
|---|---|---|---|
| #1. Was using time-in a positive experience? | 4.31 | 0.96 | 2 to 5 |
| #2. How difficult was it to use? | 2.11 | 1.28 | 1 to 4 |
| #3. How confident were you at using the procedure? | 4.37 | 0.83 | 2 to 5 |
| #4. Did you notice changes in your child once you began using it? | 4.00 | 0.04 | 2 to 5 |
| #5. Did you notice changes in the quality of your relationship with your child after you started using it? | 3.90 | 1.04 | 1 to 5 |
| #6. Do you intend to use it in the future? | 4.63 | 0.76 | 2 to 5 |

*Notes. N* = 19 (*n* = 18 for ratings 1 and 3); 1 = no or not at all, 5 = very much or many).

Several mothers, during the debriefing interview, identified challenges associated with using the technique. Foremost, they reported that it is time consuming. A mother pointed out that when she was the only adult present and had to cook dinner and watch her children, she had to wait before she had the opportunity to use it. Another mother recognized it required a conscious effort so did not use it as much as she thought she would. A third mother observed the time burden but recognized that it got easier over time, in part because her child's need for the physical connection decreased over the two weeks. One mother mentioned a family system consideration: her partner did not like the technique because he had "less patience to use tranquil strategies."

Three mothers commented on the fact that the child's reaction to time-in changed over time. Two mothers found it to be effective with less effort: it got to a point where "just a touch will be enough" to calm the child. Another mother reported how much her child liked the technique " . . . he wanted time-in versus other consequences." One astute mother observed that the effectiveness of the technique was a function of her child's mood and behavior: "Time-in is less effective when my child is really mad and more effective with emotions like frustration, sadness, and disappointment, and with sibling conflict."

## 4. Discussion

Psychologists and parenting educators have long suggested that time-in may be an effective positive parenting technique. In contrast to a controlling style of childrearing, it is designed to promote communication, connection, and respect between the parent and child. Surprisingly, although the technique has been mentioned in parenting literature, it has not been subject to empirical investigation, with the exception of two case studies.

According to the mothers in this community sample of typically developing children, time-in was an easy procedure to learn and use. Moreover, mothers found using it to be a positive experience for teaching and managing their children, and they intended to continue using it in the future. Perhaps most importantly, mothers reported that it improved their children's behavior and even enhanced their relationships with their 3- to 5-year-old children. Several mothers mentioned it prompted them to think more about their child-rearing practices and modify some of them. These findings align with the theoretical views of Alfred Adler [32] that highlight the importance of connectedness, a sense of belonging, and respect for the child's individuality and needs.

Time-in can be considered the prototypical example of positive parenting practices because it captures the essence of a positive parenting approach: focusing on the child's emotions and needs, helping the child learn how to self-regulate, and communicating openly. However, time-in and other positive parenting techniques (such as natural conse-

quences and avoiding use of rewards as well as punishments) are not well known in the general population [44]. Only one mother in our sample reported she had heard of time-in.

Positive parenting is a concept that is more than a collection of parenting techniques [36,37]. It requires a child-rearing orientation that is more child centered than parent centered with the constant goal of promoting a close, positive relationship rather than a focus on controlling the child's behavior. According to Adler, and what was hinted at in the results of this study, good child behavior will follow once a positive relationship becomes the focus. Given time-in reflects a different theoretical orientation to parenting, it conflicts with many adults' own experiences as children and contemporary child-rearing norms. As such, there will likely be resistance to adopting it among some parents.

It should be recognized that this study does not contribute to the discussion about the pros and cons of time-out [25,27,45]. Instead, our aim was to test the feasibility of training parents in using time-in and generate some initial data about its efficacy from mothers' perspectives. As we suggest below, studies are needed that directly compare the effectiveness of time-in to time-out.

Although this study yielded some positive results regarding time-in, this study has some notable limitations. Foremost, our sample of mothers was small and a self-selected convenience sample. The sample size precluded analysis of child or mother characteristics that may be related to the efficacy of the technique. For example, we did not analyze the role of precipitating child behavior to systematically analyze whether the technique worked better in response to some situations or others, as one mother observed. Another important limitation was that the data relied on self-reports. Although we sought to engage mothers as co-investigators in the study with a desire to hear both pros and cons about their experiences, they may have provided overly positive comments. It should be remembered that we developed and then trained mothers on our own version of time-in; there is no standardized, or widely accepted procedure. Variations in the design and presentation of the procedure may impact its efficacy. Another limitation is we did not assess the degree to which mothers adhered to the time-in protocol we provided, which may have introduced noise into our findings. Additionally, we collected maternal reports for only two weeks. Longer prospective studies are needed to better test out the effects of time-in.

New research efforts into time-in are needed to address the limitations in our pilot effort. Future studies should use large and diverse sample of mothers and fathers. The age of the child and type of misbehavior engaged in are two important independent variables that should be systematically studied in future investigations to inform the generalizability of our findings. In addition, assessing the effectiveness over a longer time frame, conducting in-person assessments of the effectiveness of time-in, and including multiple informants are necessary for a more definitive study. Additionally, to directly compare time-in with time-out and evaluate their relative effectiveness, carefully designed randomized controlled studies are needed.

## 5. Conclusions

The purpose of this study was to train a community sample of mothers with preschoolers to use time-in, a prominent positive parenting technique, and assess their reactions. We found that with just 90 min of instructions, the parents felt sufficiently confident to use the method and evaluated it positively. Specifically, mothers reported positive experiences using it and observed behavioral changes in their child and the quality of their interactions with their child. Perhaps most revealing, the mothers intended to continue using the procedure. Although this study is a pilot effort, with a small sample of mothers, the preliminary indications are that time-in is a promising teaching technique and merits more investigation.

**Author Contributions:** G.W.H., conception, design, and drafting; T.G. and S.E.W., acquisition, analysis, interpretation of data, and drafting; R.G., acquisition and drafting; R.A., design, acquisition and drafting. All authors have read and agreed to the published version of the manuscript.

**Funding:** Funded by an internal grant from SMU.

**Institutional Review Board Statement:** This study was approved by the SMU IRB committee. Protocol #H17-070-HOLG—Evaluating "Time-In" as a Child Management Technique.

**Informed Consent Statement:** Informed consent was obtained from all the mothers involved in this study.

**Data Availability Statement:** Contact the first author for access to the data.

**Acknowledgments:** We thank Margaret Smith for her effort in developing the time-in intervention and meeting with families. Thanks also go to Shelby Fry and Sara Stahl for their help in recruiting participants and collecting data. Last, we thank Robert Hampson, Ph.D. for supervising the time-in educational sessions.

**Conflicts of Interest:** The authors declare no conflict of interest.

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
