# Peer review of "Is It Time for “Time-In”?: A Pilot Test of the Child-Rearing Technique"

_pediatrrep, doi:10.3390/pediatric14020032_

Round 1
Reviewer 1 Report
I congratulate the authors for the article. I find it interesting and appropriate.
I believe that the concept of positive parenting practices should be further defined, beyond indicating the ambiguity of the term. One could even go more deeply into the theoretical frameworks on which it is based.
I believe that the conclusion detracts greatly from the quality of the rest of the article, so I recommend further development, not only highlighting the results of the research, but also the authors' reflections.
Reviewer 2 Report
Thank you very much for the opportunity to review your paper.
An outdated but popularly used ‘time-out’ method was very well explored, and replaced with a more novel and a contemporary approach to behaviour redirection and guidance of young children. When we have social media and popular culture such as ‘The Supernanny’ televised show where adult caregivers are easily adopting methods of the ‘time-out’ approach to assist children’s behaviours, it was worrisome that adult caregivers would still adopt this method. Thus, your research offering an alternative approach that is age appropriate, was very welcoming to read.
Some comments and suggestions are as follows:
48-49, 52: More references are required to support these arguments.
105: “Child compliance” should be made more explicit as this appears to have a quite a negative connotation. For example, readers may misunderstand that it is a positive factor that a child is “being compliant to the adult’s orders”. Thus, the clarification should be made as to how and why the children were more “compliant”. Or, a recommended terminology that is more developmentally appropriate in this age group as a suggestion, would be that ‘the child was more ‘responsive’ to XYZ…’.
173: Did the parents feel as though they had sufficient training to confidently put these strategies into practice?
299: More clarification as to how the variants of “anti-social” behaviours could be noted, and when “time-in” method was used. For example, was the “time-in” method more effective for more ‘mild’ behaviours such as refusing to share or turn-take or for more aggressive behaviours such as hitting? More examples could be noted to clarify these points and wordings such as “sibling conflict”.
I thoroughly enjoyed reading the authors’ contributions to an alternative approach to supporting children to understand their behaviours. I hope that this study can be replicated in early childhood educational settings with child-educator dyads also.
Reviewer 3 Report
The manuscript entitled, “Is it Time for ‘Time-In’?: A Pilot Test of the Disciplinary Technique” by Holden et al., is a preliminary study conducted for two weeks to assess the outcome of the “Time-in” protocol they developed intending to educate the parents about the positive reinforcement for a period of two weeks and observed an improved parent-child relationship.
Specific suggestions to improve the manuscript:
- It would be meaningful to have the ‘Time-In’ in the title within an apostrophe since this paper is about ‘time-in’ changing the ‘time-out’ concept every parent is familiar with.
- Although this study is a preliminary study conducted to educate the parents to use their “Time-in” protocol, the description lacks the aspects that are essential for the protocol. Considering there are various factors that govern the child’s behavior, including sibling rivalry, and restrictive instructions, it would be more effective if the authors describe their approach clearly.
- Pg3 ln107-109: Revise this sentence: “The present study represents an extension of existing case studies by using a non-clinical sample to assess parents’ willingness to use time-in as well as their reactions to, and perceptions of the efficacy of the disciplinary technique.” This sentence is ambiguous: the authors need to clarify what is the extension of existing and what non-clinical sample used to assess? Are they referring to non-clinical abstract methods? If so, the sentence has to be modified to convey it correctly.
- Pg3 ln124: “Nine of the children were girls but the sex of one child was unknown.” – the sex of one child is unknown – is the data on gender missing? If so, revise the sentence to indicate how many were boys and how many were girls, then specify one is missing data.
- Pg4 ln 153: “The frequency of problems scale is based on a 1= never to 7 = always rating” – revise this sentence.
- Pg4 ln165-166: “Mothers were recruited from posters displayed in several daycare centers and pre-schools in a large metropolitan southwestern American city.” – revise this sentence and provide the American city in parenthesis.
- Pg 5 ln238: responses o their children’s misdeeds – correct this sentence.
- It would be informative to present the demographic data presented in participants in a table. And, the data described in the results section (Pg5) can be presented in a table – (the numbers, SD, and significance) categorizing each parameter.
- While ‘time-in’ is positive reinforcement for the child and the parents to develop tolerance, and understanding to build a secured trustworthy relationship, in order to implementing this method, requires a cultural change. It would be informative if the authors elaborate on this and the challenges because their preliminary study is for a very short duration and the outcome of the response should be convincingly described.
Reviewer 4 Report
Authors present leur experience, acceptance by mothers and a pilot evaluation with the educational method Time-In. The manuscript describes very well the history and the interest of this method on the one hand, and the results of a standardized utilisation with a sample of mothers of typically developing children on the other hand.
The manuscript is very well written, and of great interest to the broad readership of Pediatric Reports. It also helps to sensitize readers on the interest and acceptance by parents of positif educational interventions such as time-in.
Author Response
We thank the reviewer for being so positive about our study. We are glad you liked it.

Round 2
Reviewer 1 Report
Thank you very much for your efforts.
Regards.